# Preclinical Development of Autologous Hematopoietic Stem Cell-Based Gene Therapy for Immune Deficiencies: A Journey from Mouse Cage to Bed Side

**DOI:** 10.3390/pharmaceutics12060549

**Published:** 2020-06-13

**Authors:** Laura Garcia-Perez, Anita Ordas, Kirsten Canté-Barrett, Pauline Meij, Karin Pike-Overzet, Arjan Lankester, Frank J. T. Staal

**Affiliations:** 1Department of Immunohematology and Blood Transfusion, Leiden University Medical Center, 2333ZA Leiden, The Netherlands; l.garcia@lumc.nl (L.G.-P.); A.K.Ordas@lumc.nl (A.O.); k.cante@lumc.nl (K.C.-B.); k.pike-overzet@lumc.nl (K.P.-O.); 2Department of Clinical Pharmacy and Toxicology, Leiden University Medical Center, 2333ZA Leiden, The Netherlands; P.Meij@lumc.nl; 3Willem-Alexander Children’s Hospital, Leiden University Medical Center, 2333ZA Leiden, The Netherlands; a.lankester@lumc.nl

**Keywords:** gene therapy, immunodeficiency, HSC, vector design, animal model, efficacy, safety, scaling up, regulations, GMP complaint

## Abstract

Recent clinical trials using patient’s own corrected hematopoietic stem cells (HSCs), such as for primary immunodeficiencies (Adenosine deaminase (ADA) deficiency, X-linked Severe Combined Immunodeficiency (SCID), X-linked chronic granulomatous disease (CGD), Wiskott–Aldrich Syndrome (WAS)), have yielded promising results in the clinic; endorsing gene therapy to become standard therapy for a number of diseases. However, the journey to achieve such a successful therapy is not easy, and several challenges have to be overcome. In this review, we will address several different challenges in the development of gene therapy for immune deficiencies using our own experience with Recombinase-activating gene 1 (RAG1) SCID as an example. We will discuss product development (targeting of the therapeutic cells and choice of a suitable vector and delivery method), the proof-of-concept (in vitro and in vivo efficacy, toxicology, and safety), and the final release steps to the clinic (scaling up, good manufacturing practice (GMP) procedures/protocols and regulatory hurdles).

## 1. Introduction

Over the past 5 years, the gene therapy product market has substantially expanded. Several products have been approved by the FDA (U.S. Food and Drug Administration Agency) and the EMA (European Medicines Agency) and have been granted market authorization. Among them, in 2016, the EMA approved the first ex-vivo gene therapy product using autologous hematopoietic stem cells, Strimvelis (GlaxoSmithKline), for the treatment of Adenosine deaminase (ADA) deficiency. Moreover, over 3000 clinical trials have been reported worldwide [1], with the majority addressing human cancer (CAR-T cells) and inherited monogenic diseases like primary immunodeficiencies [2]. Clinical trials with both self-inactivating (SIN) gamma-retroviruses and SIN-lentiviruses (10% of clinical trials) are currently ongoing for various primary immunodeficiencies like ADA Severe Combined Immunodeficiency (SCID) [3,4,5], X-linked SCID [6,7,8,9], Artemis SCID [10,11,12], Wiskott–Aldrich Syndrome (WAS) [13,14,15] or X-linked chronic granulomatous disease (CGD) [16,17] (Table 1). Severe combined immunodeficiencies (SCIDs) are a group of rare inherited disorders in which both the humoral and cell-mediated immunities fail to function. SCIDs are characterized by the absence of T and often B and/or NK cells, and represent a real pediatric emergency. Indeed, if not properly treated, SCIDs lead to infants’ failure to thrive associated with severe and recurrent infections and other metabolic abnormalities that are invariably fatal. Mutations in a number of genes can cause SCID: The IL2R-gamma gene mutations cause X-linked SCID; mutations in Adenosine Deaminase ADA-SCID and mutations in either of the Recombinase Activating Genes RAG-SCID. Other immune disorders include Wiskott Aldrich syndrome that also affects platelets and granulomatous disease, which affects mature granulocyte function with severe and recurrent infections and other metabolic abnormalities that are invariably fatal. Gene therapy provides a life-long cure and has the potential to become a standard clinical procedure for immunodeficiencies and some other diseases when proven safe. However, the journey to accomplish clinical trials has been extensive and laborious.

The bases of gene therapy were established following the scientific advances during the 1960s and early 1970s. Friedmann suggested good exogenous DNA could be used to replace the defective DNA in patients with genetic defects who suffer from its associated rare diseases [18]. Since then, massive efforts from basic science, translational, and clinical research have been made, reaching the first in vivo animal model evidence in 1989 that the procedure could work for primary immunodeficiencies [19]. In parallel, better understanding and characterization of the targeted diseases and improvement of laboratory methods allowed a rapid advance of gene therapy development with remarkable results for several immunodeficiencies. Although allogeneic hematopoietic stem cell (HSC) transplantation remains the prevailing therapeutic treatment for immunodeficiencies, the outcome differs depending on the source of the donor HSC (Human leukocyte antigen (HLA)–matched related donor or HLA-mismatched donor), the disease genotype, the use of conditioning, the age and the health status of the patient at the time of the treatment [20,21,22,23,24,25,26]. Despite all improvements, Graft-versus-Host disease (GvHD) remains a significant complication associated with allogeneic HSC transplantation. Therefore, particularly patients without HLA-matched donors and those with serious comorbidities would benefit from autologous gene therapy [27].

Developing successful gene therapy is not easy and several challenges need to be overcome along the journey, starting with the selection of the most suitable target cells and how to isolate them. Next, a suitable clinically applicable vector with the promoter and transgene of interest needs to be designed. Once the vector and the delivery methods have been developed, efficacy is tested both in vitro and in vivo, confirming sufficient transgene transduction, therapeutic expression of the transgene, and immune reconstitution. Furthermore, extensive toxicology and safety studies are essential to minimize potential insertional mutagenesis and clonal outgrowth due to the semirandom integration of the vector into the DNA; potentially causing leukemias or lymphomas.

The FDA [50] and EMA [51] have published guidelines that define scientific principles and provide guidance for the pre-clinical development and evaluation of gene therapy products, focusing on the quality, efficacy, and safety requirements. Extensive pre-clinical data strengthen the proof-of-concept of the potential of the developed gene therapy product, paving the way for the approval of clinical trials. However, the final release steps towards the clinic and patient treatment are lengthy due to the need for adequate scaling-up of the vector production and gene therapy product manufacturing, as well as handling mandatory regulations. From start to finish, all steps and challenges of the gene therapy development procedure (illustrated in Figure 1) will be discussed using our own experience in the development of gene therapy for Recombinase-activating gene 1 (RAG1) SCID as an example [52,53,54]. 

## 2. Product Development

Gene therapy can be broadly divided into two groups: In vivo and ex-vivo gene therapy depending on the target disease and the delivery method. In in vivo gene therapy, the transgene is administered intravenously into the patient, either by a viral or non-viral vector, and reaches the target cells inside the body. In contrast, ex-vivo gene therapy is performed outside the body. The cells of interest are isolated, modified with the therapeutic transgene, and corrected cells are transplanted back into the patient. In this last approach, the gene therapy product, i.e., the medicine, consists of the combination of the targeted cells and the therapeutic vector target cells of interest.

### 2.1. Target Cells of Interest

An essential feature of gene therapy is the persistent long-term correction of the disease, lasting for life with a unique one-time treatment, offering a cure for the disease. Moreover, primary immunodeficiencies can affect one or multiple cell lineages. In RAG1 deficient patients, both B and T cells are affected. To achieve the desired correction, a proper understanding of stem cell biology became critical as stem cells have the unique capacity of both self-renewal as well as pluripotency. Regarding immunodeficiencies, hematopoietic stem cells (HSC) are the relevant target cells that differentiate to produce all mature blood cell types for life.

Murine HSCs were first described by Becker et al. (1963) [55], describing the clonal origin of a population of hematopoietic cells. At the beginning of the 1990s, Jordan and Lemischka [56], proposed a model where single stem cell clones are sufficient to maintain lifetime hematopoiesis in an animal model and suggested that the hallmark of the long-term reconstitution system may arise from mono- or oligoclonality. Suitable markers to characterize stem cell subpopulations were identified, allowing the purification of the murine cells of interest [57,58]. The most widely-known murine HSC population is the LSK population, standing for lack of lineage markers (B220, Mac-1, Gr-1, CD3, CD4, CD8, and Ter119), and the presence of Sca1 and c-Kit. LSK comprises a heterogeneous population with different subpopulations distinguished as long-term (Thy1^lo^ Lin^−^ Sca1^+^ cKit^+^ CD38^+^ CD34^−/lo^ Slam^+^) and short-term (Thy1^lo^ Lin^−^ Sca1^+^ cKit^+^ CD38^+^ CD34^+^ Slam^−^) populations [59,60]. Mouse bone marrow (BM) HSCs can be isolated and purified by immunomagnetic beads for lineage depletion, from which a lineage negative bulk population including progenitors and long-term stem cells are collected, or by cell sorting from which a purer HSC population can be isolated. Isolated murine HSCs can then be ex vivo cultured and transduced with the therapeutic vector, followed by in vitro or in vivo testing.

In parallel, human HSCs were also identified. Weissman and co-workers (1992) [61] isolated a candidate population in human fetal BM (Thy1^+^ Lin^−^ CD34^+^) that was enriched for the clonogenic activity that established long-term and multilineage capacity. CD34 is the main marker to define human HSCs, consisting of a bulk of populations that represents around 1% of total BM cells. HSCs have been further phenotypically redefined as CD34^+^CD38^−^ cells [62] and further divided into subpopulations based on the expression of CD90/Thy1 and CD45RA [63] and CD49f. Therefore, the first full phenotypic definition of human HSCs proposed by the laboratory of John Dick (2011) [64] was CD34^+^CD38^−^CD45RA^−^CD90^+^CD49f^+^, where single defined HSCs were highly efficient in generating long-term multilineage grafts in NOD scid gamma (NSG) mouse models. Recently, new HSC markers have been identified like EPCR/CD201, which is also fairly reliable to detect HSCs in culture [65].

In the clinical setting, the CD34^+^ bulk fraction, also known as hematopoietic stem and progenitor cells (HSPC), with a mix of progenitor and long-term populations (similarly to LSK in the murine setting) is used for transplantation or gene therapy manipulation. The main advantage of using the total CD34^+^ population is the easy accessibility of these cells [66,67]. HSPCs can be harvested from the bone marrow by direct puncture or nowadays, preferably by leukapheresis. HSPCs are mobilized with two mobilizing agents (G-CSF and Plerixafor) from the bone marrow to the peripheral blood that is then collected, containing an enriched portion of HSCs [68]. HSC mobilization is a less invasive method that is routinely performed, allowing to harvest a high amount of HSPCs, also suitable for babies who are the target population of gene therapy for immunodeficiencies. Moreover, human HSPCs can easily be collected with immunomagnetic beads for CD34 enrichment, also available under Good Manufacturing Practice (GMP) compliance. Finally, HSPCs can efficiently be re-administered by infusion, where HSCs will naturally home to the bone marrow. However, there are two main challenges: Obtain a sufficient number of cells for ex vivo manipulation and successive transplantation, and achieve appropriate gene correction for cell therapy (discussed below). With regards to the need for sufficient therapeutic cells, it should be noted that HSPCs are delicate cells, and, therefore, cell loss needs to be considered during the processing of the cells (enrichment, culture, transduction, freezing, and thawing) [66]. Nowadays, achieving a therapeutic number of CD34^+^ cells is accessible thanks to the improved protocols for the collection (re-collection if needed) and isolation of HSPCs. Another way to overcome this challenge is by achieving ex vivo expansion of HSCs. Enormous efforts, as reviewed by Tajer et al. (2019) [69] and others [70,71,72,73], have been put into improving HSC culture protocols to successfully maintain and even expand the cells of interest ex vivo, and, therefore, help to overcome the shortage of primary material.

The gene therapy field is continuously evolving, offering an alternative approach by further narrowing the isolation of HSPCs to a purer CD34^+^CD38^−^ population with a clinically relevant method. A GMP compliant platform based on immunomagnetic-based cell sorting has been developed to purify large cell numbers of CD34^+^CD38^−^ cells, quickly and with high recovery [74]. This CD34^+^CD38^−^ population is more enriched with long-term HSCs, decreasing the amount of cells needed to be transduced ex vivo and transplanted back into the patient; reducing the amount of therapeutic virus needed [75]. However, myeloid reconstitution after purified CD34^+^CD38^−^ transplantation was delayed, as the first wave of immune reconstitution is known to be accomplished by progenitor cells present in the bulk CD34^+^ cells [76]. Therefore, even though a more extensive purification can improve transduction efficiency and reduce the usage of the therapeutic virus (potentially reducing therapy cost), the presence of a mixed HSPC population, including progenitor cells, is actually an advantage for a satisfactory post-gene therapy recovery.

### 2.2. Vector Design: Balancing Insertion Site and Therapeutic Expression

An optimal vector for gene therapy should carry a high DNA load capacity, enable high transduction efficiency, possess favorable cell tropism for the target cell type of interest, induce low genotoxicity and cytotoxicity, and evoke no or a limited immune response. To achieve these characteristics in the wide number of potential diseases targeted with gene therapy, a variety of vectors have been developed and optimized that can be divided into two main vector categories; non-integrative and integrative vectors. The non-integrative vectors have a safer profile, including both viral vectors such as adenoviral or adeno-associated viral vectors and non-viral vectors, which offer extra advantages on the low induced immunogenicity and the ease to produce [77]. However, as the transgene of interest will not be integrated into the DNA host cell, transgene expression might not always be stable; the expression will be retained for a prolonged period in post-mitotic tissues but diluted progressively in proliferating cells. Therefore, the application of non-integrative vectors in the hematopoietic system is limited. On the other hand, integrative vectors (mainly retroviral and lentiviral vectors) have been used in approximately ¼ of the total gene therapy clinical trials [78]. Retroviruses can enter the host cell and reverse transcribe their RNA genome into DNA that subsequently integrates into the cell DNA. As the transgene of interest stably integrates into the host DNA, a long-lasting therapeutic effect is achieved, allowing the transmission of the therapeutic material to all progeny of the transduced cells (i.e., all blood lineages developed from transduced HSCs). The use of integrating vectors in gene therapy for immunodeficiencies has a long history by now, with over 2 decades of experience since the first clinical trials started for X-linked SCID [6]. The firsts attempts of gene therapy for ADA and X-linked SCID were accomplished with a retroviral vector derived from Murine Leukemia Virus (MLV_gamma-retrovirus). Although successful correction of the disease was observed in most of the patients and no problems were observed in the ADA trial, safety issues resulted from the X-linked trials as leukemia cases appeared in patients treated with the gene therapy product. These leukemias were caused by insertion mutagenesis of the therapeutic vector. Even though retroviral integration across the DNA was thought to be random, it became apparent that there was some preference near transcriptional active sites such as oncogenes [79,80,81]. These adverse events revealed a need to develop a new generation of safer vectors with a decreased risk of insertional mutagenesis. Self-inactivating (SIN) vectors lacking potent enhancers in the LTRs were developed, for both gamma-retroviral and lentiviral vectors, however, SIN-γRV reached low transduction efficiency and expression [82]. The interest in lentiviral vectors increased thanks to their capacity to also transduce non-dividing cells and therefore allowing an increased transduction efficiency of HSCs [83]. Lentiviral vectors used in gene therapy are HIV derived and modified to guarantee vector safety.

Naldini and colleagues (1998) [84] developed the well-known 3rd generation lentiviral vector (LV) system resulting in the generation of replication-deficient LV to prevent repackaging. SIN-LV are generated based on a 4 plasmid system in which all non-essential viral genes have been removed, and the essential viral genes have been separated into several plasmids. The system consists of a group of separate plasmids: Two packaging plasmids (gag/pol and rev), a plasmid encoding for the envelope (env plasmid), and a minimal transfer plasmid with the LTRs, packaging signals, internal promoter, and the therapeutic transgene. Additionally, 3′LTRs regions were modified, resulting in the deletion of the viral promoter and enhancer activity in 3′LTR [85], and rendering the virus SIN after integration. In addition, insulators can be added into the transfer plasmid blocking the interaction between the integrating vector and the cell’s genome. Additional improvements have been implemented on the LVs aiming to enhance transgene expression and stability, thereby also increasing safety as fewer integrations are needed to achieve the therapeutic effect; fewer integrations reduce the risk as insertion mutagenesis increases with the total amount of integrations. Polyadenylation signals help to improve the correct transcript termination, improving the 3′ processing [86]. Additionally, the woodchuck hepatitis virus post-transcriptional regulatory element (WPRE) positioned behind the transgene increases RNA stability and subsequent viral titer and transgene expression [87]. Codon optimization can be an extra modification that leads to further improvement of the titer and expression by depleting secondary RNA structures and improving codon usage. Finally, as the SIN lentiviral transfer vector is devoid of LTR activity, an internal promoter needs to be included. To reduce the risk of integration in non-target tissue, the choice of tissue-specific promoters is advisable when possible. Thanks to all the advances made in vector design, SIN lentiviral vectors are the safest to date with a highly reduced genotoxicity compared to γ-retroviral vectors [85,88,89].

In parallel to the SIN lentiviral vector development, new vectors for immunodeficiencies not yet-treated, have started to be developed, such as for RAG1-SCID. As RAG1-SCID is a primary immunodeficiency, SIN lentiviral vectors were chosen for their ability to transduce HSCs and safety profile. RAG1 gene therapy development started to be developed after γRV safety issues were raised while SIN LV was continuously being improved. Therefore, a SIN LV with the native RAG1 transgene was developed, and its efficacy was evaluated. Both the in vitro (virus production and transduction) and in vivo therapeutic effect were assessed. Unfortunately, RAG1 expression was insufficient, and, therefore, a codon-optimized version of RAG1 was used (c.o.RAG1). Transduction efficiency, transgene expression, and in vivo efficacy were improved, as shown by Pike-Overzet et al. (2011) [52]. Gene therapy to treat RAG1-SCID seemed to be possible with SIN LV; however, the vector used for proof-of-concept studies was still inappropriate. Accordingly, the vector was updated into clinically applicable vectors in which different promoters were tested to achieve optimal transgene expression. Efficacy of these clinically applicable LVs was re-assessed in vitro and in vivo, revealing SIN LV MND-c.o.RAG1 as the most promising vector to correct RAG1 deficiency [54].

## 3. Proof-Of-Concept

Well-conducted clinical trials are essential to establish the benefit/risk profile. To ensure the collection of reliable data in this rapidly expanding field, the FDA and EMA have published guidelines for the development of these complex therapies. The guidelines are multidisciplinary, addressing development, manufacturing, and quality control during non-clinical and clinical development. The main objective is to provide guidance in the structure and the required data to start a clinical trial application, focusing on efficacy (in vitro and in vivo) and safety.

### 3.1. Ex Vivo Manipulation: Transduction Efficiency

One of the most important release criteria for a gene therapy product is to determine transduction efficiency by means of the vector copy number (VCN), which is a measure for the number of transgene copies integrated into the DNA per target cell. The threshold selected for these values in the therapeutic product corresponds to the lowest transduction efficiency required to ensure enough modified cells and a therapeutic effect. The ability to achieve efficient gene delivery has often been described as ‘the Achilles heel of gene therapy’ [90]. Despite the accomplishment of remarkable improvements and the development of new methods (like transduction enhancers), it is still a bottleneck to translate from pre-clinical murine models to primary human cells and finally to scale up to clinical use. To achieve a reliable VCN in the potential gene therapy product, different developmental steps need to be carefully considered: The method to determine transgene transduction, suitable lentiviral titration in therapeutic cells, and proper adjustment of the viral dose to achieve reliable and sufficient therapeutic effect.

Polymerase Chain Reaction (PCR) has been widely used to determine VCN and transgene expression in the gene therapy field for immunodeficiencies. In short, transduced cells are kept in culture for several days to avoid the detection of free plasmids and ensure the readout of stable vector integration. DNA from cultured cells is isolated to determine VCN by PCR. Sastry et al. (2002) [91] developed and established Real Time-PCR as the method for detecting LV sequences relative to a housekeeping gene. Therefore, the number of vectors inserted in the DNA was quantified, allowing the detection of multiple vector copies per cell (which was not possible with previous techniques like p24-ELISA [92]). No free plasmids were detected 4 days post-transduction by this method, indicating that detection of stable integration in the cell can be measured from that day. Across the literature, analysis of VCN is performed at different timepoints post-transduction form 7 days [12,93,94], 9 days [52,54], and up to 14 days [9,95,96]. It is important to note that; VCN values may differ when analyzed on different days, and while the differences can be subtle, it can hamper the comparison between trials. Importantly, this PCR-based method was originally generated in a way that could be used for a variety of LV vectors independently of the transgene, allowing to establish a standard method to detect VCN. However, there is no complete consensus between gene therapy studies on the vector region targeted for the PCR, the housekeeping gene, or the standard, leading to potential misinterpretations of the VCN across different laboratories and studies. Recently, the use of the Droplet Digital PCR system (ddPCR) instead of the RT-PCR has added an extra confounder to VCN determination. In principle, ddPCR offers a more accurate and reproducible detection of VCN, with minimum variability for low VCN values. Even though ddPCR is also based on the detection of a vector sequence relative to a housekeeping gene, the detection approach is different in that the VCN is calculated based on a mathematical model by Poison statistics [97,98,99].

Importantly, a precise estimation of average VCN in the targeted cell is key in defining the therapeutic product. As multi-center clinical trials are getting more attention, there is a need for more standardized protocols to define VCN’s in gene therapy products (such as the standard cell culture protocols, vector region, and housekeeping gene used) and to obtain more reliable and comparable VCN values. This is of high importance in multi-center studies where there might be a need to agree on a release VCN value across countries and different regulatory agencies. Furthermore, it would be interesting to introduce new techniques for further characterization of the therapeutic gene product such as the abundant heterogeneity regarding stem cell subpopulations and the actual percentage of transduced cells (by flow cytometry or colony-forming assays); these are key features for the success of the gene therapy outcome that is currently being assessed differently in different trials [9,12,37,54,96].

A crucial aspect for ex vivo transduction of stem cells with a lentiviral virus is the accurate determination of the viral titer of the produced viral supernatant; notably the functional titer, i.e., the lentivirus’ ability to transduce a particular cell type or cell line under specific conditions. Indeed, gene therapy is based on the most suitable amount of virus added to the target cells to obtain sufficient (but not too high) and reliable vector integration into the genome. Therefore, an accurate viral titer assessment on proper, informative cells is needed. Generally, the functional titer for different therapeutic purposes has been determined in various cell lines like HEK293T cells [91,92,95,100], Hela cells [91], HT1080 cells [101], or HT-29 cells [96]. An essential aspect to consider is that different cell lines have different permissiveness to lentiviral transduction, and, therefore, the assessed viral titer can vary depending on the cell line used; one same viral batch may have different viral titers depending on the transduced cell line. Moreover, primary cells, i.e., HSCs, are known to be more challenging to transduce. Therefore, the titer determined with a cell line may not be suitable for primary cells, getting unexpected efficiencies in the primary cells. As changes in target cell type and transduction conditions can have a dramatic effect on transduction efficiency, titration of the virus on primary cells, mainly murine HSPCs and human HSPCs from different sources (cord blood, bone marrow, and/or mobilized peripheral blood) is highly advisable. Thereby, having a specific titer in primary cells will help to achieve more reliable VCN in pre-clinical studies and in the gene therapy products across patients. Not only target cells but also transduction culture conditions should be taken into consideration, giving a more accurate read out for transduction efficiency.

As mentioned above, human primary cells (such as HSCs) can be more challenging to properly transduce and to achieve sufficient therapeutic efficiency. Together with the costly production of a clinical therapeutic lentiviral batch, transduction enhancers (TEs) are valuable compounds in the past years to boost VCN in primary HCSs using less virus if possible. Successful use of TEs will allow treating more patients with one viral batch, which will help to implement gene therapy as a standard protocol. In the past, to get a sufficient proportion of gene-corrected cells in the therapeutic product, high vector doses (2 transduction hits) and prolonged ex vivo culture (3 days in total) were needed [12,96]. Various TE compounds can be added to the culture media to increase lentiviral transduction efficiency, VCN, and transgene expression; they include Cyclosporin and Rapamycin [95], Prostaglandin E2 [93], Staurosporine [94], or LentiBOOST^TM^ [96,102]. Combinatorial TE application has also been tested, yielding even more potent effects [94,95,96]. Higher transduction due to TEs was achieved in all HSPCs subpopulation, including the long-term repopulating HSCs, without changing viability, integration sites pattern, global gene expression profiles, in vivo toxicity, or differentiation capacity in vitro (colony-forming assay) and in vivo (NSG mouse model). TEs have been tested in both murine and human cells, as well as healthy and patient donor cells, and are already manufactured in a GMP-compliant manner, facilitating their implementation in clinical protocols. In addition, TEs compounds may allow getting reliable effects to achieve the correct VCN in the gene therapy product across patients and diseases. With this approach, the use of LVs can be maximized, requiring less virus per product and enabling the use of one batch for multiple patients. For example, Schott et al. (2019) [96] showed the combinatorial use of protamine sulfate and LentiBOOST^TM^ that allows to adjust their clinical protocol by reducing the amount of virus needed and shortening the culturing time (from 2 hits strategy to 1 hit), preserving at least similar transduction. Accordingly, gene therapy for Artemis-SCID, for which preclinical studies described a 2 hits approach [12], may benefit from a similar strategy to enable adaptation to a more efficient protocol. Overall, this strategy makes gene therapy more accessible due to reduced production costs.

### 3.2. Call for Suitable Models to Test the Efficacy of Gene Therapy

The British statistician George Box stated: “Essentially all models are wrong, but some models are useful”, which can also be generally applied to scientific research models. The therapeutic effect in the gene therapy field has been demonstrated in relevant in vivo studies using a broad range of animal models from mouse to Rhesus Macaques, including dogs and pigs. However, some of these models are still far from humans, which can limit the translatability of the discoveries in non-human animals to clinical applications [103], potentially leading to failure in phase I/II clinical trials. As animal models are essential in the pre-clinical assessment, it is important to choose the most suitable disease-specific model and understand its limitations.

#### 3.2.1. Animal Models

Large animals such as dogs, pigs, and non-human primates, have been used in gene therapy [104] for several neuromuscular disorders such as myopathies, Duchene dystrophy, or Huntington [105,106,107,108], lysosome storage disorders [109,110], eye diseases [111,112], or cystic fibrosis [113]. These large models have been used to assess efficacy, dosage, route of administration, and safety. Although large animal models for immunodeficiency have been described [114,115,116], immunodeficient mice are still the most used preclinical models to study gene therapy for immunodeficiencies like SCID. Due to the broad range of available immunocompromised mouse models, including the “humanized” mouse model, human patient cells can be xenografted and directly tested in vivo. Moreover, different genetic mouse models have been developed to mimic the different forms of SCID described in humans, such as ADA-SCID [19,117,118], X-linked (IL2rg)-SCID [9,119], Artemis-SCID [10,11,12] or RAG1/2-SCID [120,121]. Importantly, these mice present a similar immunodeficient phenotype as found in humans, such as the Rag1-deficient mouse model, which present a full block at the early stages of T (DN2) and B (pre-B) cell development allowing close monitoring of the effects of the gene therapy in their development. Additionally, RAG1 and RAG2 hypomorphic SCID models are also available [122,123,124,125], allowing to study gene therapy in a wider range of immunodeficiencies with one same strategy. For example, analyzing whether the same vector can be used to correct both full RAG1-SCID and hypomorphic RAG1-SCID. Unfortunately, other SCID mouse models, such as X-linked (IL7r)-SCID does not reproduce the human setting as the mouse model has an extra B cell block that is not observed in humans [126]. With the development of new editing tools (zinc-finger nucleases, TALENs, CRISPR-Cas9), transgenic mice can be generated to reproduce SCID phenotypes that do not have an established animal model yet.

Even though we can find useful mouse models to study the efficacy of the developed gene therapy for immunodeficiencies, the gap between the mouse and the human physiological and pathological mechanisms is still substantial. The most recent achievement to overcome this gap is the development of “humanized mouse models”; immunodeficient mice such as nude or NSG mice carrying functioning human genes, human cells, or human tissues/organs. Importantly, these immunodeficient mice allow engraftment of functional human immune cells [127,128], enabling refined modeling of many areas of human biology and disease, especially immunology. This model allows sustained engraftment of human CD34^+^ stem cells isolated from cord blood, bone marrow, or mobilized peripheral blood in adult mice, developing high levels of functional lymphoid (T and B cells) and myeloid cells [129]. Humanized mice are becoming an established model to study different human diseases such as infectious diseases, cancer, autoimmunity, and testing human-specific drugs [130,131]. In the field of primary immunodeficiencies, this xenograft mouse model has allowed to provide previously unattainable insight into human T-cell development and contributes to functionally identify the arrest in thymic development caused by the three major types of SCID, as this data was largely missing due to the non-availability of thymic biopsies [132]. Although the humanized mouse model is suitable to recapitulate most of the human SCID phenotypes, murine enzymes can complement and overcome human deficiency in SCIDs that result from lacking certain metabolic enzymes. Indeed, human T, B, and NK developed from ADA-SCID CD34^+^ patient cells (T-B-NK- diagnosed patient) transplanted into NSG mice [132], in which the secreted murine ADA complemented the human deficiency, comparable to ADA enzyme replacement therapy.

Accordingly, humanized mice are an appropriate tool to study the therapeutically modified stem cells from SCID patients directly. This model has a big impact on assessing gene therapy potential in pre-clinical studies. It allows us to directly assess gene therapy efficacy and safety in developing functional immune cells. An increasing number of pre-clinical immunodeficiency studies include a proof-of-concept in patient cells transduced with the therapeutic vector and transplanted into NSG mice, to get extra therapeutic evidence closer to the human setting, hoping for a successful subsequent clinical trial. In our example, transplantation of hypomorphic RAG1-SCID patient stem cells into NSG mouse showed that functional human T cells developed after ex vivo gene therapy with our MND-c.o.RAG1 SIN LV, restoring human T cell receptor rearrangements. Such data provides additional robust evidence for starting a phase I/II clinical trial for gene therapy as an alternative curative treatment for RAG1-SCID patients [54].

#### 3.2.2. In Vitro Models

While the humanized mouse model is the best model available for pre-clinical immunodeficiency studies and irreplaceable in the foreseeable future, there is an increased pressure to reduce the number of animals used in experiments in many countries (3 R’s concept [133]). To reduce the number of experimental animals, the development of useful in vitro models is crucial. Available in vitro systems to study gene therapy in immunodeficiencies are mainly focused on T cell development. Fetal Thymus Organ Culture (FTOC) is a powerful 3D system where stroma interactions are maintained to sustain human T cell development; however, progenitor seeding efficiency and cell yield is limited, and the procedure is highly technically challenging [134]. A promising 2-D in vitro system has been developed by Zuñiga-Pflucker and colleagues where B, T and NK cell development can be studied: The OP9 co-culture system [135]. While B and NK cells develop on OP9 stromal cells, T cells need the expression of Delta-Ligand 1 and therefore develop only in the optimized OP9-DL1 system (in which B cell development is hampered) [136]. Several efforts to improve T-cell development in vitro have been performed, allowing to define an optimal medium and cytokine cocktail for optimized T cell development through all the differentiation stages up to complete single positive CD4 and CD8 T cells [137]. The OP9-DL1 system is an efficient tool for pre-clinical validation for gene therapy in cells from γc deficient patient for the correction of T cell development, as was described by Six et al. (2011) [137]. However, it can mainly be applied for T-B+NK+ SCID phenotypes. The in vitro study of more complex SCID phenotypes remains challenging. Moreover, the OP9-DL1 system cannot mediate all aspects of selection because proper positive selection of mature CD4 T cells is absent due to the lack of proper MHC class II expression [138]. Furthermore, this assay is very sensitive to subtle differences in cytokines and labile contents of culture media, making it a delicate assay [132,137]. However, these last hurdles have been overcome by the generation of an artificial thymic organoid system based on a stroma cell line expressing DL1 that efficiently initiates and sustains normal stages of T cell development from human stem cells, enhancing the positive selection thanks to the 3D structure and the new stroma cell line used [139].

Overall, with these in vitro assays, we still lack crucial information for gene therapy such as homing, long-term stability, biodistribution, or toxicology of the therapy, that can only be assessed in an animal model. Nevertheless, an interesting in vitro platform in development that may overcome the last issues is the body/human-on-a-chip. It is a flexible system that integrates human cell culture with microfluidics in vitro, integrating multiple tissues or organ system surrogates, providing a unique platform for measuring drug response or toxicity. Although still not completed, this system has a promising potential for rare disease research and orphan drug development and could be a good alternative to animal experiments [140,141].A more complex system that better recapitulates the intricacies of human T cell development is provided by artificial human thymic organoids that can be derived from iPSCS [142]. Such a system can also support the later stages of human T cell differentiation.

### 3.3. Safety and Toxicology Assessment for Gene Therapy

Not only in vitro and in vivo proof-of-concept, but also toxicology studies are requested before starting any gene therapy clinical trial. Accordingly, the EMA requires a risk assessment before the use of any gene therapy medicinal product for which toxicology, biodistribution, and integration studies have primary priority. Genotoxicity assays are not generally required but valuable and often included in pre-clinical work [143,144].

Toxicology and biodistribution are assessed by in vivo studies with the appropriate animal model (discussed above). To reduce the number of animals, all efficacy, pathology, and biodistribution studies can be combined in one same experiment. To assess the toxicology of the therapy, a full necropsy of the animals is performed after long-term reconstitution (over 16 weeks after transplantation) to assess potential long-term effects of the therapy. Organs are collected and subjected to macroscopic and microscopic examination to verify that organs look normal, healthy, and without harmful effects due to gene therapy. To assess toxicology in our RAG1 pre-clinical study, 28 organs were collected (form 14 mice in total, including controls) and analyzed blindly by a European board-certified pathologist. The selection of organs to be examined for gross pathology and histopathology analyses followed the applicable European and international guidelines (EMEA 1995, WHO 2005) [145]. For gross pathology, the external surface of the body, orifices, and the thoracic and abdominal cavities were examined. For histopathological examination, tissues were fixed in paraffin and analyzed by immunohistochemistry.

In parallel, pieces of the same organs were snap frozen to isolate DNA and determine the VCN in each organ. Vector biodistribution can then be assessed after long-term gene therapy; for HSC gene therapy the vector is expected to be present in all immune cells raised from HSC, but not detected in non-immune organs. In these cases, VCN is detected in immune organs like thymus, bone marrow, spleen, lymph nodes, and peripheral blood, but only present in low levels in other organs. However, some positivity may be observed in the intestine or lungs, as these organs have immune cells present. In addition, it is important not to detect VCN in the reproductive organs to avoid transmission to potential offspring. For RAG1, VCN in 16 organs was determined (for 8 gene therapy mice) after mice were perfused with PBS to decrease blood contamination and avoid false positives. Interestingly, if toxicology and biodistribution assays are done in parallel, the data generated can be supportive in explaining unexpected findings. For example, an infection in the lungs and stomach was detected in one of the mice during the necropsy. In parallel, a high VCN was detected in both organs of this mouse, which, thanks to the pathology observations, we could explain by the high abundance of recruited T cells in these organs. Therefore, the potential risks of unintended biodistribution of the vector were very low.

Another minimal requirement by the EMA before the use of gene therapy medicinal products is to perform integration studies. Integration studies examine the insertion sites in which the therapeutic DNA has landed. This assay became important because of the first-generation clinical trial for X-linked SCID. Some of the patients, unfortunately, developed T-lymphocyte acute lymphoblastic leukemia due to retroviral insertions near proto-oncogenes. Since then, continuous progress to develop a robust technique to detect integration patterns have been made. Schmidt et al. (2001) [146] described the first version of a technique (LM-PCR), allowing the characterization of multiple rare integrations in complex DNA samples. This technique was further improved to LAM-PCR [147] and nrLAM-PCR [148] allowing quantitative and qualitative measurement of clonal kinetics for pre-clinical studies and patient follow up; making it a robust method to understand vector integration pattern of new vectors and potential therapies as well as to detect possible malignancies derived from retroviral insertion.

Finally, although not mandatory, a genotoxicity study is advisable, which can be determined in vivo or in vitro. The in vivo assay is based on oncogenesis onset and follow up on a tumor prone Cdkn2^−/−^ mouse model. The readout for genotoxicity potential is the degree of tumor onset acceleration upon transplanting gene-corrected cells. Cesana et al. (2014) [149] showed that this sensitive method allows the detection of mild existing genotoxicity of SIN lentiviruses, and that insulators used in some vectors slightly reduce tumor growth. However, animal experiments with a tumor end-point as described are not required, as a powerful in vitro assay can be used to detect genotoxicity. In Vitro Immortalization (IVIM) assay [89,150] is based on the findings suggesting that insertional mutagenesis induce competitive growth advantages in vivo [151]. In short, primary bone marrow cells are transduced at high multiplicity of infection with the vector of interest and upon culturing and replating the selective outgrowth advantage of transformed cells is established, reflecting the transforming potential of insertional mutagenesis. The IVIM assay is convenient and shows good sensitivity, without requiring inducing leukemias or tumor growth in an animal model. Currently, an advanced version of the IVIM screening system is developed: “Surrogate Assay for Genotoxicity Assessment” (SAGA). This system integrates a molecular read-out, which enhances reproducibility, sensitivity, and reduces assay duration, paving the way for a better pre-clinical risk assessment of gene therapy vectors [152]. However, a common limitation between both in vivo and in vitro assays is the use of murine cells, from which the relevance in human cells can be questioned. Although the IVIM assay is useful to assess the risk of insertional mutagenesis, the cells are cultured in a myeloid-inducing differentiation medium favoring the readout of selective myeloid mutants (Evi1 and Prdm16) over potentially more relevant B or T cell mutants in the case of SCID therapy [89,153]. Moreover, it is a short-term assay (2 weeks in culture), which is not suitable as a readout for delayed onset genotoxicity that also occurs. Notably, although not required, the assay has been used across multiple pre-clinical studies of different gene therapy development for immunodeficiencies to assess the transforming potential of the newly developed lentiviral vectors such as X-linked SCID [9], Artemis SCID [12] or RAG1 SCID [54].

In the field, the genome editing approach has become an interesting potential alternative tool for gene addition therapy to reduce the risk of random integration and especially for the correction of tightly regulated genes expressed in specific times during development. Plenty of genome editing platforms have been developed (Zinc-finger nucleases, TALENS, Cas9 nucleases) to enable target gene correction under the physiological environment [154]. First attempts targeting the IL2RG gene to correct X-linked-SCID have been successful in pre-clinical studies [155,156]. However, the efficiency of gene correction highly depends on gene accessibility in HSCs, which can lead to insufficient therapeutic effect. Although genome editing represents a promising approach, translation into the clinic is in its infancy compared to gene addition therapies, which are still evolving as well.

## 4. Pharmaceutical and Clinical Development Phases: From Mouse to Human Treatment

After successful pre-clinical development, the next step in product development is the translation to a gene therapy medicinal product suitable for clinical use. Intensive labor to make the product suitable for the clinic includes scaling up, development of good manufacturing practice (GMP) compliant manufacturing, and complying with other directives and regulations.

### 4.1. Scaling Up: GMP Protocols and Manufacturing

The manufacturing of both the lentiviral starting material and the gene therapy medicinal product (CD34^+^ cells transduced with the therapeutic transgene) need to be scaled up and translated to a GMP compliant manufacturing process. The first step is to find a manufacturer for the lentiviral vector with appropriate manufacturing facilities and a GMP license; this can be an academic center or a commercial partner. When manufacturing is outsourced, good technology transfer from the research group to the manufacturer is crucial. Lentiviral production of our GMP RAG1 lentiviral clinical batch was outsourced to Batavia Biosciences B.V. Although this company has extensive expertise in virus production for clinical use, it was the first time they produced a lentiviral vector-based product upon adaptation from the research into a large scale GMP protocol. In our example, we will manufacture the gene therapy medicinal product (i.e., gene-modified CD34^+^ cells) in-house, within the academic environment of the Leiden University Medical Center (LUMC) thanks to the availability of proper GMP compliant cleanroom facilities with suitable equipment. To fulfill the gap between research and GMP production, collaboration with knowledgeable departments and qualified personnel has been essential. Furthermore, personnel were encouraged to get familiarized with the relevant regulation and a GMP working environment.

A common hurdle when stepping into GMP manufacturing (viral vector starting material or gene therapy medicinal product) is that all raw materials and disposables should be available in appropriate quality and equipment must be qualified according to GMP guidelines. A GMP compliant quality system should be in place ensuring amongst other the qualification of starting raw materials and suppliers, traceability and the qualification and validation of analytical assays and equipment. Since cell-based medicinal products cannot be sterilized after manufacturing, it is crucial to ensure an aseptic manufacturing process. This can be done by working in qualified cleanrooms (class A in B), making use of closed systems, and applying appropriate measures to avoid cross-contamination [157]. To comply with GMP guidelines is not easy, and research-based protocols may need to be adapted to the new requirements, encountering different challenges that will be discussed below.

#### 4.1.1. GMP Compliant Virus Manufacturing

GMP RAG1 lentiviral vector manufacturing was outsourced and protocols were adapted upon technology transfer from the research laboratory to the Contract Manufacturing Organization. Research protocols needed to be adjusted to produce large volumes, where viral production and concentration systems can differ, moving towards more sophisticated methods to ensure a high-quality product. As the gene therapy field is rapidly reaching several clinical trials, the production of highly concentrated and purified large scale virus batches is in demand. To achieve such manufacturing, the use of a stable lentiviral producing cell line would be ideal as cell lines are easy to scale up and adapt to serum-free medium and culture in suspension. Nevertheless, suitable GMP lentiviral producing stable cell lines are not yet available [158,159,160,161]. Therefore, the transient transfection protocol on adherent cells is used. However, it can be challenging to achieve large numbers during the upstream processing (virus production) and to reach high concentration and purity during the downstream processing [162].

The upstream process is costly as high-quality raw materials are expensive and need to be largely used. Adherent cells used for lentiviral production (HEK293T cells), as well as raw materials and plasmids, should ideally be the same as used in pre-clinical studies but at a higher quality grade in most cases. The main problem with adherent cells in the scaling-up is the huge surface area and the laborious manipulation needed. From a simple culture flask, large scale protocols are adjusted to multi-layer flasks, allowing a higher surface to culture cells in the same space. However, the increase in LV production remains modest as cell density is still “low”. Improvements to allow gene therapy to become a standard therapy and overcome the lack of scalability have been made taking different approaches. Systems that increase cell density by extending the surface to adhere have been developed, such as hollow fiber bioreactors [163] or fixed-bed bioreactors [164]. In addition, the field is moving towards adapting adherent cells to suspension cultures, achieving greater cell density and easier scaling-up.

After overcoming the challenge to produce large volumes of lentiviral supernatant, downstream processing of the sample is crucial to achieve high purity while maintaining high viral titers. Different methods are available with diverse relevant parameters concerning scalability, prices, capacity and throughput, removal of contaminants, maintenance of functional virus, and product losses. The most suitable procedure should be selected to concentrate and remove impurities, i.e., anion exchange chromatography [165,166], size exclusion chromatography [167], affinity absorption chromatography [166], or tangential flow filtration [168]; although a streamlined combination of many techniques is likely to be chosen.

Knowledge transfer when manufacturing is outsourced, and scalability issues are major challenging stages of the lentivirus production stages. The last bottleneck is the extensive LV quality control that needs to be performed on each produced batch. LV production and manufacturing are performed under EU guidelines, which requires rigorous safety controls of the LV starting material. Release tests to assess microbial contamination and purity (free of endotoxin, bacteria, yeast, mycoplasma, toxic agents and residual host cell protein and DNA), safety (Replication competent lentivirus negative and residual plasmid negative) and potency (viral titer and transgene identity assessed) of the LV starting material are recommended for routine batch analysis. Extensive characterization of purified LV GMP grade batches is needed to reduce potentially harmful effects of the therapy in the following steps.

#### 4.1.2. GMP Gene Therapy Product Manufacturing

Protocols to successfully isolate and transduce human CD34^+^ cells with the produced GMP grade LV needs to be adjusted to be able to manufacture a suitable medicinal product for clinical use under GMP compliance. CD34^+^ isolation is adjusted to use a close system purification instrument, such as CliniMACS from Milteny Biotec, which is currently a good selection system design for processing a large number of cells [169]. An updated semi-automated system, Prodigy, is also available and can successfully enrich and transduce CD34^+^ cells with minimal user manipulation [170]. With this GMP compliant equipment and obtaining a comparable yield, purity, and transduction efficiency as current protocols, this semi-automated cell isolation, and transduction equipment has the potential to improve the availability and standardization of HSC gene therapy. Further adaptations in the transduction protocol of the CD34^+^ cells are needed, both due to the high number of cells that have to be transduced and to the transduction method itself. Spin-oculation is often used in the research setting [52,54] to increase transduction efficiency. However, it was decided that this approach was not suitable to be performed in the cleanroom, due to the high number of cells. The process would be too laborious, which would have a negative impact on the cells and in addition, would enhance the contamination risk as a result of an extensive open production step. Subsequently, this change in the protocol promotes the use of alternative high-quality grade available transduction enhancer methods such as TEs compounds discussed above.

When facilities, personnel, documentation, raw materials, and equipment are ready, validation runs have to be performed, which will show whether the gene therapy medicinal products can be generated successfully, reliably, and aseptically. In the process and release test, such as determining cell numbers, viability, transduction efficiency, and product sterility, should be ready at this point too. It is also very important that the development of analytical assays start early in product development. If the medicinal product is given as a fresh product to the patients, then this constitutes the final product. If the product is cryopreserved, thawing protocols need to be evaluated, as well as post thawing viability of the product. Stability testing of the fresh or frozen product, as well as the lentiviral starting material, is also required and have to be considered during the development process and optimally is combined with the validation runs. Once protocols have been adjusted for clinical use and validation runs have been fully accomplished, one can proceed with the final clinical applicable protocol.

### 4.2. Regulatory Hurdles

Developers of gene therapy products not only face challenges in the scientific and technological fields but also experience additional hurdles in the regulatory trajectory, even though the regulatory environment for ATMPs (Advanced Therapy Medicinal Products) has been globally coevolving with the increasing interest in marketing authorization over the past decade [171,172].

The regulatory requirements may be complex and vary across continents and countries, but their aim is always the same: To ensure the safety and well-being of human beings [173]. The three major regulatory authorities in the European Union (EU), the United States (USA), and Japan have been making great efforts to develop and implement tools that facilitate ATMP development and enable products to reach the patients as early as possible. They have been making important steps to define appropriate regulatory standards; however, due to the novelty of this field and the complexity of such products, regulators face scientific issues never discussed before. As a result of this, regulatory requirements for approval for market authorization are not standardized or harmonized yet [172,174].

In the EU, the legal framework for ATMPs is laid down in the European Regulation (EC) No. 1394/2007, known as the ATMP regulation, amending Directive 2001/83/ EC and Regulation (EC) No. 726/2004. This regulation is in place since 2009. It defines ATMPs and ensures that such products are subject to appropriate regulatory evaluation before their clinical and commercial use, according to the regulatory framework for human medicinal products [175]. Numerous other directives apply to clinical gene therapy. An overview of the European legislation, legal guidelines, and guidance on various relevant subjects with regards to ATMPs can be found at the website of the EMA [176].

Before an ATMP can be granted with market authorization and be available for patients, clinical trials have to be performed to demonstrate safety and efficacy. To conduct clinical trials, sponsors should follow specific requirements to obtain national authorizations from the regulatory body of the individual EU member states [173]. The review process to assess the benefit-to-risk ratio for patients is currently regulated by Directive 2001/20/EC. This lengthy process, currently 90 days, has been widely criticized by the scientific community. Due to their call for reform, the EC will replace this directive with the new ‘Regulation 536/2014 on clinical trials on medical products for human use’. It is expected to come into effect in 2020 [177,178]. According to this new EU regulation, clinical gene therapy is still considered as a special case, and the review period for gene therapy products can be extended by an additional 50 days [177].

During the process of obtaining regulatory authorization, the sponsor must submit a clinical trial application (CTA), also known as a standard research file, to the national competent authority and an independent ethics committee. The research file includes several essential documents: Clinical trial application form, trial protocol/amendment(s), written informed consent form(s), subject recruitment procedures (e.g., advertisements), written information for subjects, Investigator’s Brochure (IB), Investigational Medicinal Product Dossier (IMPD), summary of scientific advice, available safety information, information about payments and compensation available to subjects, the investigator’s current curriculum vitae and/or other documentation evidencing qualifications, and any other documents that the regulatory body may need for the review.

Besides, in EU countries, separate legislation has been implemented to assess the environmental risks of genetically modified organisms (GMOs) within clinical gene therapy trials. Thus, in addition to approval by an independent ethics committee and the competent authority, GMO license must also be obtained before trials can commence. The additional GMO legislation is based on two environmental EU directives, the 2009 EU directive entitled ‘Contained use 2009/41/EC’ and the 2001 EU directive entitled ‘Deliberate release in the environment 2001/18/EC’. Unfortunately, there are several issues related to the environmental risk assessment process. First, these directives failed to keep up with the scientific progress and gene therapy vector development of the past 25 years and still apply to the current clinical gene therapy trials of which the environmental risks can be considered negligible [177]. Second, this process does not only require longer review timelines but are also poorly harmonized within the EU. While in the Netherlands, the deliberate release framework always applies, resulting in a lengthy procedure, in the UK, the length of the procedure depends on the biological characteristics and environmental risk assessment of the GMO. To reduce the review timeline, integration of environmental risk assessment in the EU clinical trial legislation framework was put into the consideration of the EU Parliament’s and European Committee. Furthermore, it was suggested that only one organization should be considered responsible and accountable for the review of clinical gene therapy trials similar to the USA. Such improvements would result in a more efficient and transparent review process and reduce the time needed for the product to reach the patients [177]. Overall, harmonization of GMO authorizations across the European would clearly facilitate clinical trials with GMOs.

If a trial has a multinational design involving more EU Member States, there is a possibility to make use of the voluntary harmonized procedure (VHP). In 2009, VHP was introduced by the Clinical Trials Facilitation Group as a pilot of Regulation 536/2014. The objectives of the VHP are to establish harmonized assessments and decisions on clinical trials in the EU and identify possible serious issues before the official submission [179]. The possibility of obtaining centralized approval for participating member states could facilitate the approval procedure for the study in a timely fashion. The VHP takes place prior to the official national submissions of the research file. Documents like the protocol, the IB, and IMPD are assessed jointly by one participating member state and the other concerned member states of a VHP. Although this procedure is an efficient tool to achieve harmonized and quick approvals of clinical trials in many EU Member States in one procedure, it currently has no formal status. It is an informal procedure that does not lead to an official decision. The submitting party can, therefore, not derive any rights from the procedure, and sometimes regulatory approval by national agencies can be delayed due to conflicts in the VHP rules with national requirements. However, as there also are clear benefits to having a VHP approval, sponsors should certainly consider pursuing this route.

It should be noted that a large team effort is indispensable to compile a research file that can be granted regulatory approval. Continuous collaboration of preclinical developers, clinicians, pharmaceutical, legal and health technology assessment experts, project managers, patient organizations, and regulatory experts is required [173,180]. Although universities are a major player in the field of ATMP development, unfortunately, researchers face significant hurdles partly due to lack of regulatory expertise and related financial support [180,181]. Thus, next to working with the necessary specialists, early engagement with the national competent authorities and/or EMA is encouraged to succeed in development. University researchers are advised to discuss scientific questions via scientific advice at the national regulators, as this is often easier, cheaper, and can be relatively informal or via the EMA’s Innovation Task Force. This way, regulators could provide scientific advice to ensure the development plans are acceptable and in line with regulatory expectations. Furthermore, with their contribution, the potential for lengthy and costly delays can be reduced [173,182].

The application is approved, and the trial can commence, but that does not mean the work is done. During and after the trial, sponsors must follow relevant regulatory regulations and frequently submit different information to the regulatory authorities (e.g., amendments, SAEs, SUSARs, SADEs and line listings, progress reports, DSU report, a summary on trial results).

When the whole development trajectory is performed successfully, one can apply for market authorization (MA). For ATMPs a centralized marketing authorization is mandatory, which leads to a single marketing authorization that is valid in all EU countries. The EMA, together with its Committee for Advanced Therapies (CAT), Committee for Human Medicinal Products (CHMP), and the network of national agencies, are responsible for the scientific evaluation of the MA applications [183]. CAT offers an ATMP classification and high-level expertise to assess the quality, safety, and efficacy of ATMPs. It reviews whether the clinical development and product manufacturing processes comply with the particular standards and requirements and ensure that the data presented are complete, accurate, and satisfactory [175]. To facilitate the authorization process, EMA provides early access tools and support. The priority medicines (PRIME) scheme is the main tool. PRIME was introduced in 2016 to enhance support for products targeting an unmet medical need and to speed up evaluation, thus, the medicines can reach patients earlier [171]. It is interesting to note that over one-third of the medicines in the PRIME scheme are ATMPs and all of these are gene therapies. In the United States and Japan such support schemes are also actively contributing to the progression of cell and gene therapies [175]. The EU has also released guidelines supporting a risk-based approach to cover quality, safety, efficacy, manufacturing, and biological aspects. The risk-based approach is a strategy to determine the level of data required and to support justification for any deviations made from directives [173].

Unfortunately, ATMPs are often seen as products with a low commercial value and/or a high commercial risk due to the complex manufacturing processes, orphan indications, and tailored production [180]. Currently, all three regulatory authorities show a willingness to accept uncertainty and safety risks with the emphasis of paying accurate attention to post-marketing surveillance and risk-minimization measures [174,180]. As of May 2019, 14 ATMPs have been granted a MA for the European Economic Area, however, 4 of them already have been withdrawn from the market for a variety of reasons [180]. An alternative route to increase access of drugs to patients in Europe is through the hospital exemption, which allows the use of ATMPs under the supervision of a medical practitioner, on a non-routine basis, and in restricted circumstances, in a single member state.

In summary, the development, regulation, and clinical use of most ATMPs are constantly co-evolving, and this path should be further followed. Although the use of gene therapy products and lentiviral vectors, in particular, is still relatively new for developers and regulators, the regulatory authorities can provide guidance and useful information on quality, safety, and efficacy during product development. In the coming years, the number of authorized gene therapy medicinal products is expected to increase significantly. When more information on such products is available, regulations and guidance are expected to increase and be harmonized, thereby supporting the delivery of more promising new gene therapy medicinal products in the EU and global markets [173].

## 5. Conclusions

Ex-vivo gene therapy using HSCs has extensively progressed over the last three decades, allowing to establish a relevant workflow of the essential steps to develop gene therapy from the pre-clinical stage to clinical trials [184,185]. Obtaining robust pre-clinical data to initiate a successful clinical trial is laborious, time-consuming, and with the risk of becoming a vicious cycle. Every time an improvement in the vector design needs to be implemented or protocols (isolation or transduction) are adapted, in vitro and in vivo efficacy should be tested again. Additional safety tests might also be required, entering successive cycles of adaptation and improvements. Although HSC isolation procedures and used vectors are rather standardized in the field, it is important to keep basic research in parallel to their clinical use, allowing continuous optimization in GMP compliant manufacturing and automated procedures as well as improving vector safety. A key parameter in the generation of the gene therapy product during clinical implementation is the ex-vivo manipulation procedure, which is currently lacking standardized assays—this lack of standardization results in variability between assays and inconsistency between research groups and developing therapies. Standardization of viral titer and VCN determination will help to overcome the variability in one of the most important release criteria of the gene therapy product. Data from in vivo and safety studies are crucial to initiate the way into the clinic. Approved and reliable in vitro safety tests are used regularly. However, a significant amount of the toxicology and safety results are obtained from experiments performed in animal models, revealing the importance of choosing suitable models for the diseases. Once favorable pre-clinical data following the FDA and EMA guidelines is collected, it is time to step out of the pre-clinical development cycle and step towards the pharmaceutical and clinical development phases, with still an extended journey ahead until starting a clinical trial. As gene therapy remains an emerging treatment, GMP manufacturing and regulations have been developed in parallel to the clinical implementation progress made in gene therapy. Communication between researchers, industrial representatives, and regulators is key to learn and grow together in this new field, adapting as the therapy evolves to design solid guidelines for the standard implementation of gene therapy as a medicinal treatment. Although the focus of this review has been on autologous HSC-based gene therapy for immune deficiencies, similar approaches are being used successfully for red blood cell disorders (such as thalassemia) and a wide range of metabolic disorders affecting brain, liver, and muscle (reviewed by Staal, Aiuti and Cavazzana [186]). In all these diseases, a long path of development (Figure 2), starting with suitable vectors and disease-specific mouse models, was required to reach clinical implementation.

## Figures and Tables

**Figure 1 pharmaceutics-12-00549-f001:**
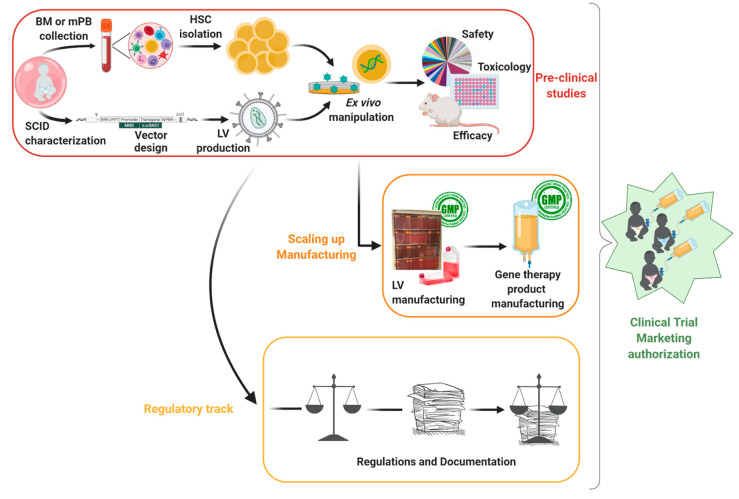
Overview of the pre-clinical assessments of gene therapy treatment: From disease modeling to clinical application (Bone Marrow (BM); mobilized Peripheral Blood (mPB); Hematopoietic Stem Cell (HSC); Lentiviral Vector (LV); Severe Combined Immunodeficiency (SCID)).

**Figure 2 pharmaceutics-12-00549-f002:**
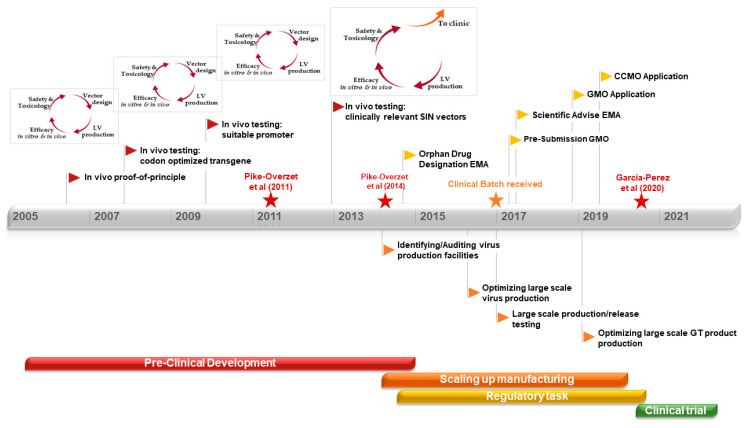
Development of autologous stem cell-based gene therapy for RAG1 severe combined immuodeficiency (SCID): A journey from mouse house to bed side. (Lentiviral Vector (LV); Centrale Commissie Mensgebonden Onderzoek (CCMO); European Medicines Agency (EMA); Genetically Modified Organism (GMO); Gene Therapy (GT)).

**Table 1 pharmaceutics-12-00549-t001:** Summary of finished and ongoing clinical trials for primary immunodeficiencies.

Disease	Gene	Vector		Clinical Study	Phase	Participants	Location	Study Type	Status	Outcome/References
**ADA-SCID**	ADA	RV		NCT01279720	Phase 1/2	8	EU	interventional	completed	Positive benefit-risk profile [28,29,30,31,32,33,34,35,36]
				NCT00018018	Phase 1	8	USA	interventional	completed	
				NCT00794508	Phase 2	10	USA	interventional	completed	
				NCT00599781	Phase 1/2	8	EU	interventional	completed	
		Strimvelis (RV)		NCT00598481	Phase 2	18	EU	interventional	completed	Immune reconstitution [33,37]
				NCT03232203		10	EU	observational	recruiting	
				NCT03478670		50	EU	observational	enrolling	
		SIN LV		NCT01852071	Phase 1/2	20	USA	interventional	completed	Immune reconstitution, well tolerated [36,38,39]
				NCT02999984	Phase 1/2	10	USA	interventional	completed	
				NCT01380990	Phase 1/2	36	EU	interventional	completed	
				NCT04140539	Phase 2/3	3	USA	interventional	recruiting	
				NCT03645460	n.a.	10	China	interventional	recruiting	
				NCT04049084		70	USA/EU	observational	enrolling	
**Artemis-SCID**	DCLRE1C	SIN LV		NCT03538899	Phase 1/2	15	USA	interventional	recruiting	Immune reconstitution [40]
**Chronic granulomatous disease**	Gp91 phox	RV		NCT00927134	Phase 1/2	2	EU	interventional	completed	Sustained engraftment, insertional mutagenesis, [41,42]
				NCT00564759	Phase 1/2	2	EU	interventional	unknown	
			SIN	NCT01906541	Phase 1/2	5	EU	interventional	unknown	
				NCT00778882	Phase 1/2	2	Korea	interventional	active	
		SIN LV		NCT01855685	Phase 1/2	3	EU	interventional	active	[16]
				NCT02757911	Phase 1/2	3	EU	interventional	active	
				NCT02234934	Phase 1/2	10	USA	interventional	active	
				NCT03645486	n.a.	10	China	Interventional	active	
**Leukocyte Adhesion Deficiency-I**	CD18	RV		NCT00023010	Phase 1	2	USA	observational	completed	
		SIN LV		NCT03812263	Phase 1/2	9	USA	interventional	recruiting	
				NCT03825783	Phase 1	2	EU	interventional	recruiting	
**Wiskott–Aldrich Syndrome**	WAS	SIN LV		NCT01347242	Phase 1/2	6	EU	interventional	completed	Successful engraftment, immune reconstitution, no adverse reactions [15,43,44]
				NCT01347346	Phase 1/2	5	EU	interventional	completed	
				NCT02333760	Phase 1/2	10	EU	interventional	active	
				NCT03837483	Phase 2	6	EU	interventional	active	
				NCT01410825	Phase 1/2	5	USA	interventional	active	
				NCT01515462	Phase 1/2	8	EU	interventional	active	
**X linked-SCID**	IL2RG	RV		NCT00028236	Phase 1	3	USA	interventional	completed	Sustained immune correction, risk acute leukemia [6,45,46,47,48]
			SIN	NCT01175239	n.a.	1	EU	interventional	unknown	
				NCT01410019	Phase 1/2	5	EU	interventional	unknown	
			SIN	NCT01129544	Phase 1/2	8	USA	interventional	active	
		SIN LV		NCT03315078	Phase 1/2	13	USA	interventional	recruiting	Multilineage engraftment, immune reconstitution [49]
				NCT03311503	Phase 1/2	10	USA	interventional	recruiting	
				NCT01512888	Phase 1/2	28	USA	interventional	recruiting	
				NCT01306019	Phase 1/2	30	USA	interventional	recruiting	
				NCT03601286	Phase 1	5	EU	interventional	recruiting	
				NCT04286815	n.a.	10	China	interventional	recruiting	
				NCT03217617	Phase 1/2	10	China	interventional	recruiting	

Adenosine Deaminase (ADA); Severe Combined Immunodeficiency (SCID); Retroviral Vector (RV); Lentiviral Vector (LV); Self-Inactivating (SIN); Not Applicable (n.a.).

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
