# Peer review of "Preclinical Development of Autologous Hematopoietic Stem Cell-Based Gene Therapy for Immune Deficiencies: A Journey from Mouse Cage to Bed Side"

_pharmaceutics, 2020, doi:10.3390/pharmaceutics12060549_

Round 1
Reviewer 1 Report
The authors reviewed the essential aspects and stages involved in ex-vivo gene therapy using HSCs for the treatment of primary immune deficiencies. They lucidly summarized the completed and ongoing clinical trials, discussed the aspects of product development of ex-vivo gene therapy using HSCs including the cells, vectors, and animal models. The authors talked in details about the steps from the research to clinic and presented an important perspective on the difficulties for this important therapeutic.
Overall, the manuscript is well written. My minor comments are listed as below.
- Table 1, It would be beneficial if the outcomes of the clinical trials could be reviewed and described briefly. Alternatively, it would be worthy to review the current status of HSC-based gene therapy for each disease, and/or add the relevant references.
- The reviewer suggests adding a cut-off date for the presented numbers, for example, page 2, lining 37, ‘over 2000 clinical trials have been reported worldwide’. There are now more than 3000 clinical trials have been reported as time of Sep 2019 (http://www.abedia.com/wiley/index.html).
- Line 84, Has any clinical trial used SIN vector?”
- Please define the abbreviations in the abstract.
- Please fix typos, such as line 156.
Author Response
- Table 1, It would be beneficial if the outcomes of the clinical trials could be reviewed and described briefly. Alternatively, it would be worthy to review the current status of HSC-based gene therapy for each disease, and/or add the relevant references.
Thank you for indicating this important missing information. As suggested, the outcome and relevant references on the outcomes of the clinical trials have been added and can be found in a new column of table 1 (Outcome/references).
- The reviewer suggests adding a cut-off date for the presented numbers, for example, page 2, lining 37, ‘over 2000 clinical trials have been reported worldwide’. There are now more than 3000 clinical trials have been reported as time of Sep 2019 (http://www.abedia.com/wiley/index.html).
Thank you for this remark. Indeed, total number of clinical trials is over 3000 (3025) nowadays. This information have been updated as follow: line 51 “Moreover, over 3000 clinical trials have been reported worldwide [1]”.
Line 84, Has any clinical trial used SIN vector?”  
Yes, SIN lentiviral vectors are being tested in different clinical trials for WAS, X-linked and ADA SCID. To clarify this point, this information has been added to the table 1 and in line 53 “Clinical trials with both self-inactivating (SIN) gamma-retroviruses and SIN-lentiviruses (10% of clinical trials) are currently ongoing”.
- Please define the abbreviations in the abstract.
The abbreviations from the abstract have been defined following your advice.
Line 24-26 “such as for primary immunodeficiencies (Adenosine deaminase (ADA) deficiency, X-linked Severe Combined Immunodeficiency (SCID), X-linked chronic granulomatous disease (CGD), Wiskott–Aldrich Syndrome (WAS))” and Line 30 “with Recombinase-activating gene 1 (RAG1) SCID as an example”. Besides, a new list of abbreviation has been included in the manuscript (page 2).
- Please fix typos, such as line 156.
Typos have been revised along the manuscript, and can be traced with the track changes.
Reviewer 2 Report
The subject of the review is interesting and actual. The authors describe all the steps required to use stem cell based gene therapy in immunodeficiencies.
Major revisions:
The primary immunedeficiencies cited in the paper by Garcia-Perez et al. are ADA-SCID, X-SCID, CGD, WAS, RAG1-SCID. The authors assume that the all the readers know these pathologies, but as this paper is a review it will be useful to add a paragraph with a brief description of the diseases.
Line 358: “development of “humanized mouse models”. Please briefly describe the characteristics of this animal model.
Minor revisions:
Line 34: the authors indicate the meaning of the acronym EMA “European Medicine Agency”, actually the meaning of EMA is European Medicines Agency. Please correct
Lines 628-629: why the authors explain EMA again?
Line 611: I couldn’t find the explanation of the acronym ATMPs. Please add.
Line 120: “long-term multilineage grafts in NSG mouse models”. Is NSG the acronym for “NOD scid gamma mouse”?
Line 613: “he regulatory requirements may be complex”. Please change “The regulatory requirements may be complex”
Line 684: “In should be noted that a large team effort is indispensable to compile a research file”. Please change “It should be noted that a large team effort is indispensable to compile a research file”
Author Response
The primary immunedeficiencies cited in the paper by Garcia-Perez et al. are ADA-SCID, X-SCID, CGD, WAS, RAG1-SCID. The authors assume that the all the readers know these pathologies, but as this paper is a review it will be useful to add a paragraph with a brief description of the diseases.
Thank you for this remark. A brief description of the diseases has been added in the introduction from line 57 to 66 “Severe combined immunodeficiencies (SCIDs) are a group of rare inherited disorders in which both the humoral and cell-mediated immunities fail to function. SCIDs are characterized by the absence of T and often B and/or NK cells, and represent a real pediatric emergency. Indeed, if not properly treated, SCIDs lead to infants’ failure to thrive associated with severe and recurrent infections and other metabolic abnormalities that are invariably fatal. Mutations in a number of genes can cause SCID: the IL2R-gamma gene mutations cause X-linked SCID; mutations in Adenosine Deaminase ADA-SCID and mutations in either of the Recombinase Activating Genes RAG-SCID. Other immune disorders include Wiskott Aldrich syndrome that also affects platelets and granulomatous disease which affects mature granulocyte function with severe and recurrent infections and other metabolic abnormalities that are invariably fatal”
Line 358: “development of “humanized mouse models”. Please briefly describe the characteristics of this animal model.
This point has been revised as requested from Line 402 “The most recent achievement to overcome this gap is the development of “humanized mouse models”; immunodeficient mice such as nude or NSG mice carrying functioning human genes, human cells, or human tissues/organs. Importantly, these immunodeficient mice allow engraftment of functional human immune cells [127, 128], enabling refined modelling of many areas of human biology and disease, especially immunology.”
Minor revisions:
- Line 34: the authors indicate the meaning of the acronym EMA “European Medicine Agency”, actually the meaning of EMA is European Medicines Agency. Please correct
Corrected as requested (Line 48)
- Lines 628-629: why the authors explain EMA again?
This issue has been corrected as follow “to ATMPs can be found at the website of the EMA [176]” (line 691)
- Line 611: I couldn’t find the explanation of the acronym ATMPs. Please add.
The acronym has been added in line 672 “ ATMPs (Advanced Therapy Medicinal Products)”
- Line 120: “long-term multilineage grafts in NSG mouse models”. Is NSG the acronym for “NOD scid gamma mouse”?
Indeed, NSG is the acronym for NOD SCID gamma mouse. It has been clarified as follow in line 151 “HSCs were highly efficient in generating long-term multilineage grafts in NOD scid gamma (NSG) mouse models.”
- Line 613: “he regulatory requirements may be complex”. Please change “The regulatory requirements may be complex”
Done (line 675)
- Line 684: “In should be noted that a large team effort is indispensable to compile a research file”. Please change “It should be noted that a large team effort is indispensable to compile a research file”
Done (line 751)
Reviewer 3 Report
The authors summarized gene therapy using hematopoietic stem cells (HSCs) in this review, and provided information on preclinical and clinical studies of stem cell-based gene therapy. While this review covers interesting topics, there are some things to improve. Here are my suggestions:
- Figures are not easy to understand for readers. The authors need to improve the resolution of images and make the characters larger.
- This manuscript includes many abbreviations. A list of abbreviations will be helpful for the readers.
- The authors titled autologous stem cell based gene therapy, but did not mention other autologous stem cells. Therefore, the authors need to add information about other autologous stem cells or revise the title.
Author Response
- Figures are not easy to understand for readers. The authors need to improve the resolution of images and make the characters larger.
Thank you for indicating this issue. The figures have been updated with higher resolution and increased size to improve their understanding for readers (see page 5 and 20).
- This manuscript includes many abbreviations. A list of abbreviations will be helpful for the readers.
Thank you for the advice. A list of abbreviations has been added from line 42 (page 2).
- The authors titled autologous stem cell based gene therapy, but did not mention other autologous stem cells. Therefore, the authors need to add information about other autologous stem cells or revise the title.
The title has been revised and changed to “Preclinical development of autologous hematopoietic stem cell-based gene therapy for immune deficiencies: a journey from mouse cage to bed side “, as we mainly focus on hematopoietic stem cell-based gene therapy
Round 2
Reviewer 2 Report
The subject of the review is interesting and actual. The authors describe all the steps required to use stem cell based gene therapy in immunodeficiencies.
The authors increased the level of the paper by better clarifying some aspects. Also the new title of the review better summarize its content.